# Partnership towards Synergistic Municipal Solid Waste Management Services in a Coastal Tourism Sub-Region

**Surasak Jotaworn [1,*], Vilas Nitivattananon [1], Kyoko Kusakabe [2] and Wenchao Xue [3]**

1   Urban Environmental Management Program, Department of Development and Sustainability, School of Environment, Resources and Development, Asian Institute of Technology, Klong Luang, Pathumthani 12120, Thailand; vilasn@ait.asia

2   Department of Development and Sustainability, School of Environment, Resources and Development, Asian Institute of Technology, Klong Luang, Pathumthani 12120, Thailand; kyokok@ait.asia

3   Department of Energy, Environment, and Climate Change, School of Environment, Resources and Development, Asian Institute of Technology, Klong Luang, Pathumthani 12120, Thailand; wenchao@ait.asia

*   Correspondence: p_pet555@hotmail.com

**Abstract:** Solid waste generated on land could potentially contribute continuously to marine waste, with current municipal solid waste management (MSWM) focusing on human-related activities as the main source. While there has been challenges and opportunities in the MSWM's partnership in the growing waste generation for the coastal tourism area, the aim of this study is to explore public and private sectors as the key players to identify challenges, opportunities, and need for further analysis of the synergistic MSWM services in the Eastern Economic Corridor (EEC), Thailand. A mixed-method approach was adopted, including primary data collected through semi-structured interviews and questionnaire surveys. Content analysis, descriptive statistics, and chi-square tests were applied. The results show that the public sector has different MSWM strategies—with public-private partnership (PPP) and without PPP, with many challenges in the EEC region—while the private sector has a lot of potential for MSWM effectiveness. The synergistic opportunities from both sectors can therefore be considered for possible integration into four aspects: challenging synergies within the public sector, potential synergies via the private sector, synergies with a cross-sectoral partnership, and synergies through other types of partnership. Additionally, a synergic partnership was another appropriate approach for MSWM services enhancement.

**Keywords:** MSWM; coastal tourism sub-region; public and private partnership; synergistic opportunities; partnership effectiveness

## 1. Introduction

The municipal solid waste (MSW) context needed special attention because of its complex production and growing volume. If it is mismanaged, the MSW may cause air and water quality degradation, become a public health issue, contribute to climate change (e.g., releasing methane), and be the main particulate pollutant in the oceans in the shape of (micro) plastics [1]. The interconnectedness between land and coastal and marine environments is paramount to achieve successful waste management. Waste management problems in the coastal area may result in marine management problems due to the linkages between ecosystems [2]. There are many studies explicating that most marine litter originates from land-based activities [3–5].

In the coastal tourism area, municipal solid waste management (MSWM) is currently focused on developing their operations and technology based on the diverse waste composition and the waste origins from various sources [5]. There is no great concern from the relevant sectors nor collaboration for developing tourism waste management. Accommodation providers receive visitors for one or more nights; restaurants and leisure

enterprises provide food and tourists' services. These two sectors (accommodation and tourism enterprises) have become the main producer of waste in these tourist sites. To be increasingly given extended producer responsibility (EPR), the private sector along the coastal area must be aware of the impacts of their business no the environment [6].

Based on a news report about five years ago on marine waste in Thailand, the trend is dramatically increasing [7]. It is estimated that 80% of marine plastic waste comes from onshore activities such as community waste dumping sites on the shore and from beach tourism, while 20% comes from marine activities such as sea transport, fishing, and marine tourism [8]. Moreover, a report on municipal solid waste disposal sites in Thailand in 2018 announced the critical provinces of uncollected waste, i.e., Rayong (253,046 tons) and Chonburi (119,611 tons). Both provinces are located in the Eastern Economic Corridor (EEC) region. Currently, the Thai Government expects the EEC region to become the hub of the Association of South East Asian Nations (ASEAN), and there have been many collaborations for mega projects in the fields of economy, investment, and technology [9]. On the other hand, there were fewer projects and limited partnerships for environmental conservation.

To prevent further destruction of nature, the current MSW in the coastal tourism sub-region that directly impacts marine nature should be holistically considered by the relevant sector for waste generation rather than focusing on MSWM technology or human/tourist behavior only. This study, therefore, aims to explore public and private sectors to identify the challenges, potential, and need for further analysis of the synergistic opportunities towards a partnership for effective MSWM in the EEC region, a major coastal tourism destination in Thailand.

## 2. Literature Review

Based on the main objectives of the research, the literature review was structured systematically to realize the rationale of this research and the literature gaps. Coastal tourism development and waste management in tourist destinations have been made aware of the issues that require special attention because of the link between coastal growth and the ecosystem. Meanwhile, a gap was found in the challenges of a synergistic partnership as there is no synergy approach study in MSWM enhancement, especially for the coastal tourism sub-region.

### 2.1. Coastal Tourism Development

Currently, the coastal zone is impacted by urbanization, which is the expansion of cities and towns due to a significant rise in the migration of people to the urban and coastal zone. The migration trend has been forecasted as 55% in 2018 up to 68% by 2050. Asia and the Pacific regions had the most dramatic movements of urbanization in 2018. It was revealed that the Southeast Asia (SEA) population alone increased by 44% from the 1960s to 2010 and is expected to rise 60% by 2030, particularly in the coastal zone [10].

The transformation of the coastal zone is intended for economic and tourism benefits. Tourism is the main factor in the drastic rise of high population density. It provided 277 million jobs for the global economy in 2014. It is not surprising that coastal urbanization, natural capital, and local traditions can attract millions of tourists. Thus, the sub-regional areas are important for coastal tourism [11].

### 2.2. Waste Management in Tourism Destinations and the Opportunities Needed

The world's environmental pollution is a partial product of the severe problems in coastal tourist destinations [12]. Beach and marine environments have already been plagued with pollution exacerbated by the tourism industry. Even the municipality has distinguished waste management charges in tourist destinations for specific sectors such as hotels/resorts, restaurants, and tents along the beach, but this still results in inadequate waste management [10]. This indicates that local governments, such as municipalities, and the private sector have limited environmental awareness and collaboration as they

cannot find a win-win situation from those collaborations [13]. The public sector expects all stakeholders to have a high degree of understanding of the environmental effects of tourism, as opposed to the private sector, which expects all stakeholders to increase the economic potential of tourism without environmental concerns. In the sense of a win-win, the benefit derived can be both in-kind and in-cash or in the value of resources and should be shared and clarified clearly along with the processes for working together.

Due to its effect and rising volume, municipal solid waste (MSW) needs special attention. MSW offers complex compositions and origins based on the number of sources. Whether it is included in the category of solid waste or not varies dependent on the laws of each country [11]. Whatever the category it is, when it was mismanaged, it can cause air and water quality degradation, be a public health concern, and lead to climate change (for example, the release of methane). Eventually, it will become the largest particulate pollutant in the oceans in the form of (micro) plastics [14]. Hence, acknowledging the link between land and coastal and marine ecosystems is essential to the effective management of waste. To prevent adverse effects on the natural capital from rapid development in the coastal area and, at the same time, have long-term collaboration to boost the economy by tourism, synergistic opportunities are needed on solid waste management in the coastal tourist sub-region.

### 2.3. The Challenges for Synergistic Opportunities

### 2.3.1. Lack of a Synergistic Partnership Study

To handle MSW in the tourism sector, many options are needed whereby the government and policymaker can decide on the best approaches based on grounded theory. There are many MSW techniques and tools available to evaluate the quality of solid waste management [15]. Some assessments are based on mathematical tests where data on waste management are available, such as Life Cycle Evaluation Methodologies [16], Material Flow Analysis [17], and Multi-criteria Decision Analysis [18]. The use of sustainability assessment is also widespread and can be measured not only by scientists and researchers but also by city officials and local workers in the solid waste field [19,20]. Additionally, the collaboration between the different management processes (sorting, collection, transport, and destination) for MSW is of utmost importance. For sustainable MSWM development, it has been shown that a dynamic organizational transformation is required which includes multi-sectors with different resources and expertise to contribute to the operation [21]. The literature has shown the validity of this approach. However, there was a lack of any synergistic study on partnership effectiveness specifically for waste management issues in the coastal tourist sub-region. It is, therefore, necessary to add more research to the ground theory that has been compiled to identify how to achieve the sustainability of MSWM in a coastal tourism sub-region.

### 2.3.2. The Complexities of the Synergy Concept

In society, many individuals, groups, and associations exist, having different self-interests. If there was no regulatory mechanism to control each identity, chaos could be the outcome [20]. Synergy is a term that has been derived from the Greek word "Synergos" meaning working together [22]. "Working Together" is a method or process that is worked on together for the same purpose. The definition of synergy is the ability of two units or organizations to generate greater value working together than they would apart. Thus, it represents a dynamic process that involves adapting, learning, and creating an integrated solution that entails joint action from many partners in which the total effect is greater than the sum of individual effects when acting independently [23]. The Synergy Model is readily adaptable to the acute care or critical care setting when the patient is critically ill and the intensive care nurse links his or her competencies to the patient's needs [24]. It describes a framework for nursing practice. The key to this model is the linkage of a patient's characteristics with a nurse's competencies to achieve optimal patient outcomes [25]. The rational principle of this idea is based on the development of services

during peak times crowded by patients. Hence, it is reasonable to apply it to the MSWM services in tourist destinations that have a similar context to improve MSWM services along with the increasing trends of tourism.

However, there are differences in ecology cases, as the synergy concerns the combined effect of emission-reducing measures and harmful impacts which provide challenges and barriers for mitigation and adaptation [26]. Synergy should translate into greater efficiency (less effort, time, and resource)) or added value to the results. Therefore, it is combining the potential measures for reducing emissions with the potential measures for preventing the harmful impacts. For example, the practical guide of synergistic analysis for enterprise development.

Figure 1 shows the overlapping area between Business Environmental Reform (BER) approaches and Green Growth (GG) policies. This is referred to as GBER and is where possible synergies can be realized. The principles of BER and GG are compatible in their focus, despite variations in scope and goals [27]. Thus, this article is seeking synergistic opportunities for the common focus and growth in MSWM from the public and private sectors in the coastal tourism sub-region.

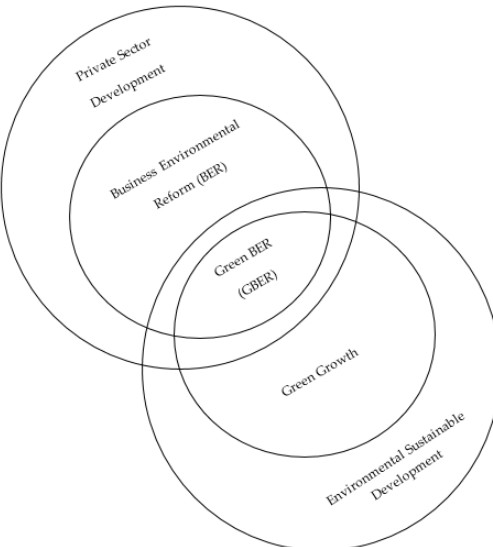

**Figure 1.** Green Business Environmental Reform Synergy [27].

This interdisciplinary research is expected to synergize the different organizations in the public and private sectors in the coastal tourism sub-region. MSWM itself has to balance the economy by creating service satisfaction for stakeholders and the environment in which waste controls need to be managed for conserving the existing nature. Therefore, the integration of the concept of synergy from different fields is inevitable in order to academically increase efficient analysis and realize the diversity of empirical findings. Besides, the private sector will be assessed by the service characteristics, consumption, attitude, and adoption of the green growth assessment [27] as its green competency for identifying the challenges, potential, and needs. The public sector will need to adapt to the sustainability assessment which is popularly used in many countries [18,19].

2.3.3. Partnership Synergy

The power to combine the perspectives, resources, and skills of a group of people and organizations has been called synergy [28]. It has been hypothesized that this distinguishing feature of collaboration is the key mechanism through which partnerships gain an advantage over single agents in addressing the system's issues. Identifying the synergy or output of a partnership will influence the effectiveness of the partnership's outcome [29]. A partnership's efficiency connotes how well it optimizes the involvement of its partners. To maximize synergy and keep the partners engaged, it needs to be efficient

to the extent to which the roles and responsibilities of partners match their interests and skills and ensure the benefits derived from being in the partnership. Thus, the partnership synergy provides an opportunity for creative, comprehensive, practical, and transformative thinking to support the synergy that results in sustainability [30].

In summary, coastal tourism development cannot deny the link between the growth of the economy and natural capital conservation. MSWM service enhancement needs special attention to keep the balance between both of these aspects. A synergy concept is an appropriate approach that has been used for over a thousand years with various meanings [31]. However, synergy is significant in a development context, which is popularly used in the health care context to serve as a leverage for effective, equitable, and sustainable growth [23]. Thus, a synergistic approach may produce outstanding results and worthwhile technical assessments for the ground theory.

## 3. Methodology

### 3.1. Research Design and Overall Methodology

As sub-regional areas are important for coastal tourism, they have impacted the coastal urbanization, natural capital, and local traditions by attracting millions of tourists to visit [10]. Thus, MSWM in coastal tourist destinations needs special attention due to the currently mismanaged waste which can cause the largest pollutants in the oceans in the form of (micro) plastics [1]. Moreover, the local government and the private sector have less environmental awareness and limited collaboration [15].

Figure 2 was developed from the Green Business Environmental Reform Synergy (Figure 1) which is summarized the body of knowledge from the global forum of DCED with 22 members, Cambridge, UK [27] by isolating the circle, changing the content, and adding the related sectors instead. Finding the challenges, potential, and needs from the two different sectors were important as baseline data to further analyze synergistic opportunities [26]. The synergy study, however, included a variety of concepts: Healthcare services [25], ecological science [26], and organizational transformation [27]. This article has to integrate all the concepts reviewed as interdisciplinary analysis to identify the maximum synergistic possibilities for enhancing MSWM services. The partnership synergy [29] was the final discussion step to ensure that the opportunities found could improve the above-mentioned problem statement and fill the gap in the literature providing another validity approach for the grounded theory and the quality of partnership effectiveness in MSWM services in the coastal tourism sub-region.

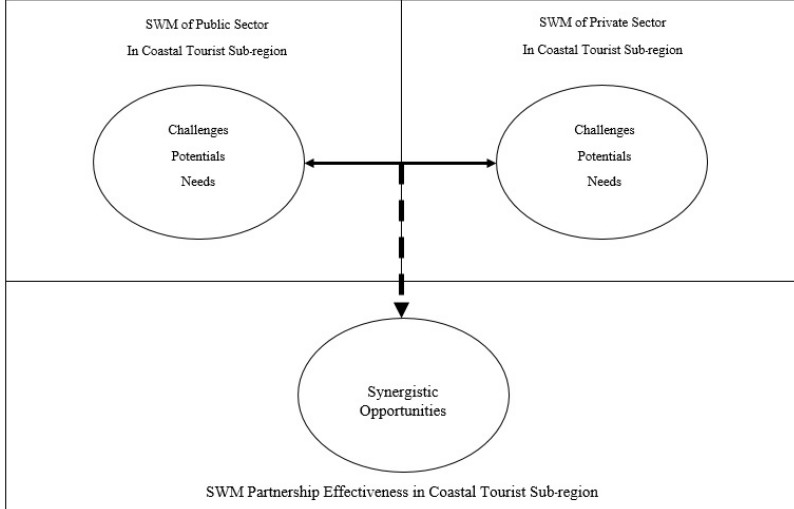

**Figure 2.** Conceptual Framework, adapted from [25,27,29].

The mix-method was adapted to explain what the synergistic opportunities are when the public and private sector partners for effective MSWM services in the coastal tourist des-

tination. Qualitative methods were used to investigate municipalities by a semi-structured interview and for tourism businesses (accommodations and trades) along the beach by a questionnaire survey.

As the main objective was to explore the public and private sector for further analysis of synergistic opportunities, municipalities were selected as the unit analysis of the public sector to adopt the sustainability assessment, while tourism businesses along the beach including accommodation (hotel and resort) and trades (restaurants, grocery store, and tents) were the unit analysis of the private sector to evaluate the service characteristics, attitude, and green competency. After the literature review was completed, an appropriate five-step study was designed to prevent the complexity of a synergistic partnership analysis in coastal tourist sub-region as follows:

- Step 1: Investigate the selected municipalities through sustainability assessment to identify the overall challenges, potentials, and needs.
- Step 2: Explore the identified private sector through a questionnaire survey to identify the overall challenges, potentials, and needs.
- Step 3: Analyze the synergistic opportunities from the public and private partnerships.
- Step 4: Discuss the findings and recommendations from those results for partnership effectiveness.
- Step 5: Conclusion based on the result of the partnership towards synergistic MSWM services in the coastal tourism sub-region, Thailand.

### 3.2. Study Areas

Thailand hopes to develop the sub-region (EEC) in the eastern part into a leading ASEAN economic zone. The EEC straddles three eastern provinces of Thailand—Chonburi, Rayong, and Chachoengsao—off the Gulf of Thailand, and the government hopes to turn those provinces into a hub of technological manufacturing and services with strong connectivity to its ASEAN neighbors. This region currently contributes 20% to Thailand's GDP and is a popular tourist destination for all. For coastal tourism development, one of the EEC's plans is to increase the growth of tourists up to 46 million arrivals which will lead to a greatly increased environmental impact [9]. Even though this region has high potential in many dimensions (e.g., technology, service, and tourism), it will also have a severe impact each year on the existing environment and natural capital.

Based on Figure 3, there are only two provinces that have many attractive coastal tourism destinations: Chonburi and Rayong. The distance of all the beaches in the two provinces is longer than 200 km. The coastal area of Chachoengsao is mostly the river delta "Bang Pa Kong" area with fertile mangrove and plenteous small animals. With 13,285 square km across the three provinces, the four beaches are shown in Figure 3, with only 20 km. long distance, can make a large impact at the regional level and also influence the economic consequences if the waste and visitors are not well-managed.

Table 1 provides basic information of the selected study area, four coastal areas have been chosen for the EEC region: Bang San, Ban Amphur, Mae Ram Phueng, and Pa Yun beaches. These areas have different beach lengths. Many tourists, both domestic and international, like to visit this region's beaches, islands, and nightlife. These coastal tourist destinations have been chosen as the study area based on the following criteria: (1) Urban tourism, (2) Located in the coastal area of the EEC region, (3) Encountered waste generation problems because of economic and tourism growth. To ensure reliability, these selected areas have complied with the above-considered criteria [32].

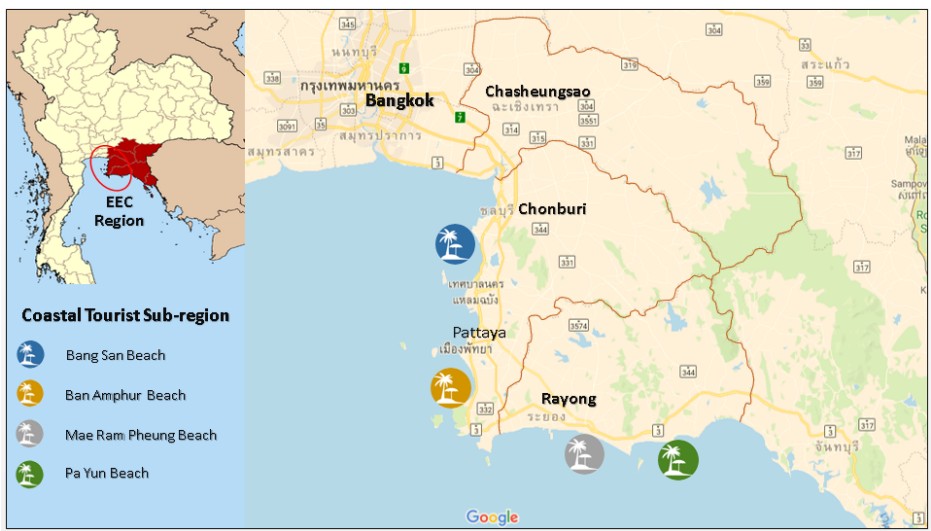

**Figure 3.** Study Sub-Region and Areas.

**Table 1.** Basic Information of the Chosen Study Areas in 2018.

| Selected Tourism Destination | Local Government | Beach Distance (km.) | Population (Residents) | Visitors in High Season (January to March) |
|---|---|---|---|---|
| Bang San Beach | San Suk Municipality | 4 | 46,368 | Chonburi 4,536,626 |
| Ban Amphur Beach | Na Chom Tian Municipality | 3 | 8189 | |
| Mae Ram Phueng Beach | Ban Pe Municipality | 6.5 | 16,392 | Rayong 1,912,119 |
| Pa Yun Beach | Ban Chang Municipality | 7 | 29,376 | |

Source: [33].

### 3.3. Data Collection

Finding the synergistic opportunities for the public and private partnerships to enhance the MSWM services in the coastal tourism sub-region needs interdisciplinary research. Both the primary and secondary data used were important to increase reliability and validity.

For primary data, the semi-structured interviews were designed to cover aspects of sustainability assessment. The head of the public health and environment division of selected municipalities must be the key informant investigated to ensure reliability. From June to July 2019, face-to-face interviews with key informants were carried out. The semi-structured interviews were transcribed and quoted similar or relevant issues from all interviewees. The checklist questions in the semi-structured interview are included in the Appendix A. The questionnaire survey was used to analyze the private sector in the coastal tourist destinations. Many stakeholders were identified in the tourism services business: Museum, trade, craft, meeting and events, sport, transport services, hotel, restaurant, university, government, tour operations, travel agency, and theme park [34]. Accommodation (hotel, resort, and small room rental) and trade (restaurant, local shop/grocery, and tents) are the key clusters that generate much of the solid waste in the tourism industry based on the EEC study area and the life cycle of tourists. These two sectors along the beach were thus the target sampling of this analysis.

There were four aspects of the survey: service characteristics, consumption behavior, attitude, and green competency which included pollution reduction, environmental protection, decreased resources, and carbon intensity. To measure the consistency of the questionnaire, Cronbach's alpha reliability test was done for the pilot test. A 95% confi-

dence level using Yamane's simplified formula of the population represented each study area [22]. The sampling technique was based on the registered private sector data provided by the municipality. The number of sample sizes for the area was calculated at 95% confidence level by using Yamane's simplified formula of the population represented for each study area [35].

Table 2 shows that the total accommodation sampling was 88 units, and the total of trades was 113 units. Thus, the total sampling of the private sector was 201 units used to generalize the coastal tourist destination in the EEC region. The way of reaching and selecting the sample was the convenience technique which was affordable to access the respondents. Mostly the sample was in the study area who met certain practical criteria, i.e., accessibility and willingness to participate. However, the accommodation managers and trade owners who allowed the collection of data represented only 198 samples from the total of private sector samples from July to November 2019.

**Table 2.** Sampling of Private Sector for Questionnaire Survey.

| Private Sector Types | Bang San | | Ban Amphur | | Ban Pe | | Phayun | |
|---|---|---|---|---|---|---|---|---|
| | Reg. No. | Sampling No. | Reg. No. | Sampling No. | Reg. No. | Sampling No. | Reg. No. | Sampling No. |
| Accommodations | 40 | 36 | 15 | 14 | 30opp | 28 | 11 | 10 |
| Trades | 50 | 44 | 23 | 21 | 35 | 32 | 17 | 16 |

Source: Semi-structured Interview, 2019.

For secondary data, the reports by the Pollution Control Department [36], municipality ordinance, the official statistics report, research articles, and related reports were utilized to obtain supporting evidence on the MSW performance of both sectors in the coastal tourism destinations.

### 3.4. Data Analysis

As this research applied a mixed-method approach, the content analysis, the descriptive statistic, and chi-square were used to analyze the synergistic opportunities toward partnership effectiveness for MSWM in the coastal tourism sub-region.

Based on the qualitative approach, the specific analytical method for the semi-structured interview was content analysis. The interview result was transcribed into the common issue and the specific critical issue to summarize the challenges, potentials, and needs for MSWM in the study area. Numbers and statistics collected from the interview were interpreted in a bar chart and table platform to compare and analyze the result findings. To ensure validity, all quoted results were checked again with the content of each aspect in the sustainability assessment.

Data analysis of the questionnaire survey of the private sector was used with the frequency, means, standard deviation, and percentage by using the SPSS program, version25.0 (available:https://www.ibm.com/analytics/spss-statistics-software). However, to find more possible synergies, the chi-square test, both association and homogeneity, were adapted to assess the relationship between green competency and private sector types.

### 4. Results

### 4.1. Public Sector Investigation: Different Strategies in MSWM

Based on the sustainability assessment, there were six aspects of performance indicators: Policy, organization, technological, social, economic, and environmental. The overview of results from the selected municipalities found that the MSWM of selected municipalities could be grouped into two types: With public-private-partnership (PPP) and without PPP. This was impacted by other aspects of performance indicators both commonly and specifically.

### 4.1.1. Policy Aspect

The decentralization in Thailand gave municipalities the autonomy to respond to the policies being transferred. All selected municipalities have different MSWM strategies in their territories: with PPP and without PPP. By assessing the different characteristics of MSWM, the common and specific challenges and potentials for both PPP and without PPP were found.

With PPP municipalities, they have divided the responsible areas into two zones, i.e., tourism and community. Municipalities were fully responsible for the tourism zone directly, while the community zone was the responsibility of the private partners. The tourism zone has been supported and collaborated with by many private companies and enterprises (e.g., sorting waste campaign, PR board, signs, and colored-bins) for marketing their organizations. For municipalities without PPP, they mostly covered the MSWM services for the whole area. Both the tourism and community zones had the same MSWM services implemented without consideration of the different volumes and compositions. Hence, without PPP, municipalities lacked the same private sector support as PPP municipalities and also faced budget limitations.

Additionally, the adequacy of waste management policies transferred to LGAs was considered sufficient coming from many government agencies. As a result, the selected municipalities had said that the current waste management policies were adequate and that decentralization was not, currently, a major problem, however, reporting on MSWM performance back (bottom-up) to the government was a major challenge in each fiscal year.

### 4.1.2. Organizational Aspects

PPP municipalities seem to have more advantages from the perspective of the policy aspect in terms of MSWM strategies and capacity. But there was a critical gap with their private partners. The partnered private organization wants to make a profit for their business, but municipalities mostly have a budget limitation. Since waste was a type of work that not many sectors wanted to do, and the amount of waste was increasing every year, municipalities needed to continually negotiate to sustain the operations and relations.

For municipalities without PPP, there were many disadvantages compared to municipalities with PPP. In terms of roles and responsibilities; those without PPP have the role of the operator to cover all MSWM processes, while those with PPP have the role of an inspector to monitor the MSWM operations. For labor tenure; without PPP municipalities do not have enough manpower and expertise, while with PPP municipalities have received the certified labor from the private partner.

### 4.1.3. Technical Aspects

There were two points related to the technical aspects; the volume of seasonal waste and the communication process among stakeholders. Based on the interviews and the secondary data review, the results showed that the recorded amount of waste between high and low seasons per day was significantly increasing. The obtained data is displayed in Figure 4.

Figure 4 shows that waste generation in the high season was double compared to that generated in the low season. San Suk and Na Chom Tian municipalities had an increasing volume of around 25–30 ton/day, while Ban Phe and Ban Chang municipalities had an increasing volume of 20–22 ton/day. The local government should pay attention to the seasonal waste, if they did not have enough equipment, trucks, and manpower for waste management in high season, it would raise the volume of uncollected waste and impact the marine environment. Meanwhile, if some local governments did not have well-planned waste management technologies, the amount of waste input in the low season would be another challenge.

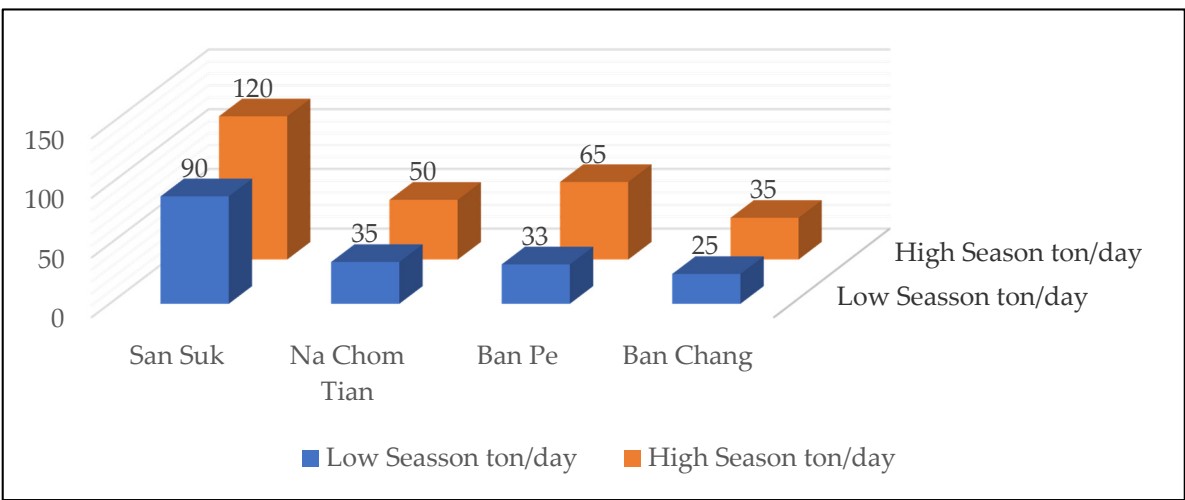

**Figure 4.** Seasonal Waste Generation in 2018 [32].

For the communication process to request MSWM participation, the semi-structured interview found that PPP municipalities mostly communicated through a private partner manager, tourism business network, or people group, while without PPP municipalities communicated directly to each enterprise and person by themselves.

### 4.1.4. Social Aspects

All selected municipalities delivered consistent opinions on stakeholders that the municipality was currently collaborating well with as a mediator with other stakeholders in society. However, even the people sector (residents) claimed that waste management was a service to them and society, but it was very difficult for visitors to participate especially the excursionist.

Table 3 shows that the trend of visitors was growing with both domestic and international types. The majority of tourist types in the EEC region were excursionists who stayed less than 24 h in the tourist area. This kind of tourist has a behavior that is quite careless regarding their consumption and produced waste. With a close distance between the EEC and Bangkok and the current improvement of transportation, most people preferred a one-day trip to the EEC rather than to other areas. Therefore, the growth of the excursionist type in the EEC region was not surprising. Thus, the challenge for all municipalities was to request tourist participation to support MSWM while other stakeholders could participate

**Table 3.** Internal Excursionists in Chonburi and Rayong Province.

| Excursionist | January–March | | |
| --- | --- | --- | --- |
| | **2018** | **2017** | **%Change** |
| Chonburi | 557,987 | 528,224 | +5.63 |
| Thai | 493,520 | 467,839 | +5.49 |
| Foreigners | 64,467 | 60,385 | +6.76 |
| Rayong | 754,482 | 698,886 | +7.95 |
| Thai | 692,784 | 639,317 | +8.36 |
| Foreigners | 61,698 | 59,569 | +3.57 |

Source [32].

### 4.1.5. Economic Aspects

The MSWM service charge collected from residents was almost the same in all areas, there was no difference in charges between the tourism and community zones. Municipalities had been using the determined charge since the municipal ordinances were introduced

in 2000 which was an equal rate. Thus, residents were quite satisfied to pay. For the tourism business, if they did not pay the MSWM service charge, they would not be permitted to continue their business nor license in the tourism zone. However, the semi-structured interview found that both citizens and the tourism businesses were afraid of the charges rising when the newly elected government transferred the policy. It was another challenge for those municipalities to keep stable tax capacity.

Figure 5 presents MSWM expenditures and shows that San Suk municipality fluctuated the trend because it relied on the number of tourists for its economy, while the MSWM expenditure for the private partner to respond in the community zone was consistent. Ban Pe municipality without a PPP group had a decreasing trend since they transferred MSWM responsibility of Samet island to the provincial level. For Na Chom Tian with a PPP group and Ban Chang municipality without a PPP had an increasing trend of MSWM expenditure. Thus, the financial viability and trend of municipalities with a PPP had more stable expenditure and fluctuates over a period, depending on the economy and the contract of the private partner, while municipalities without a PPP had gradually increased MSWM expenditure every year.

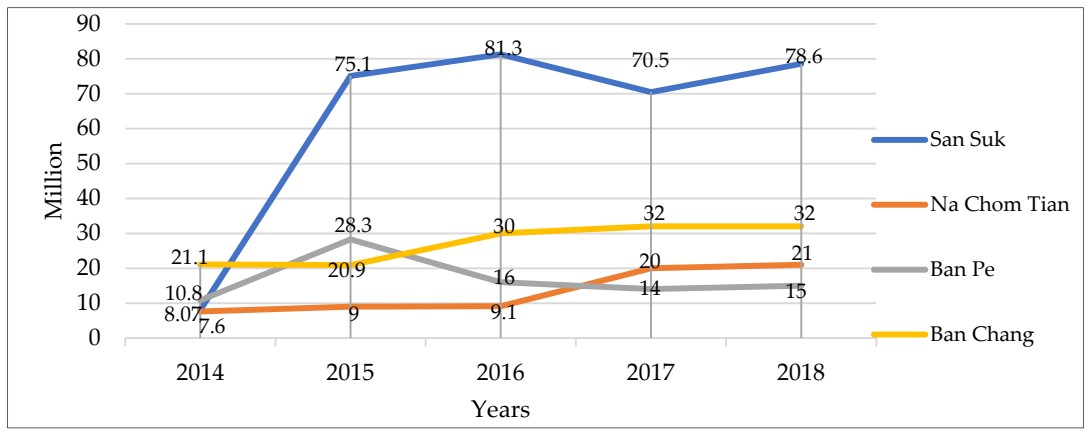

**Figure 5.** Five-year graph of MSWM expenditures [37–40].

4.1.6. Environmental Aspects

All selected municipalities indicated that the current MSWM had improved from the past due to quality development and sufficient equipment. Municipalities self-rated the cleanliness equally up to 90–95% from the current waste generation. Although there were several reports of invasion dumping within their areas, there had never been a case of illness or public health impact from those situations or from landfill.

Apart from the sustainability assessment, all selected municipalities were asked about the need to improve the MSWM capacity. Several common needs were identified from all selected municipalities which are summarized in Table 4:

**Table 4.** Summary of Public Sector Investigation Results.

| | | Challenges | | Potentials | | Needs |
|---|---|---|---|---|---|---|
| Common | PuN1 | Complexity in reporting the WM performance back to the government, | PuN1 | Sufficient policy | PuN1 | Need people to be more conscious about cleanliness and proper waste disposal. |
| | PuN2 | Changing and increasing the MSWM charge from new transferred policy | PuN2 | Full authority from the decentralization system | PuN2 | Need scholars and academia do more research on MSWM |
| | | | PuN3 | Willingness to collaborate with other sectors | PuN3 | Need NGO to run SWM activities and campaign |
| Specific | PuN3 | Negotiate to control the MSWM cost with a private partner, | PuN4 | The advantage in organizational and technical aspects of having a private partner | PuN4 | Need more PR strategies on MSWM |
| | PuN4 | The disadvantage of the role as an operator, | | | PuN5 | Need all enterprises to have green behavior |
| | PuN5 | Deficiency of manpower, | | | PuN6 | Need more CSR activities from the private sector |
| | PuN6 | Increasing MSWM expenditure of those without a PPP | | | PuN7 | Need stricter enforcement on cleanliness |
| | PuN7 | Non-efficient landfill | | | PuN8 | Need a better way of reporting back to the government |
| | PuN8 | Ineffective communication | | | | |

Note: PuC (Public Sector Challenges), PuP (Public Sector Potentials), PuN (Public Sector Needs).

Table 4 provides the challenges and potentials found as the common and specific aspects based on the indicators of the sustainability assessment. Even without a PPP, municipalities had more disadvantages in several aspects, however, the critical challenge aspect existed in municipalities with a PPP. Hence, there were opportunities to analyze the synergy internally (with the same MSWM service provider together) from an inter-governmental perspective, and externally (cross-sector) to enhance MSWM in the coastal tourist sub-region. Besides, the common needs identified from the public sector also referred to other sectors or new organizations which can be exploited later for synergistic opportunities analysis.

*4.2. Private Sector Exploration: Change Agents*

The private sector in the coastal tourism sub-region can be grouped mostly into accommodation (hotel, resort, and small hostel rental) and trade (restaurant, local shop, and tents). The questionnaire survey explored four aspects; service characteristics, consumption behavior, attitude, and green competency. It was found that the green competency of all private sectors had a positive result. Even the private sector could not deny that they were the main waste generator, but the attitude and green competency could become the potential to drive the change in the coastal tourism sub-region.

As shown in Table 5, the accommodation group mainly consisted of hotels, resorts, and others; collected data at 57.4%, 38%, and 4.6%, respectively. Other types were apartments and hostels. The trades group was comprised of restaurants, local shops, and tents,

collected at 33%, 32%, and 35% respectively. Both the business size and operational years distributed a quite similar ratio.

**Table 5.** Characteristics of the Tourism Private Sector in the Study Area. n = 198.

| Accommodation Characteristics | Accommodation | | Trades | | Trade Characteristics |
|---|---|---|---|---|---|
| | **F** | **%** | **F** | **%** | |
| Business type | | | | | |
| Hotel | 50 | 57.4 | 37 | 33 | Restaurant |
| Resort | 33 | 38 | 35 | 32 | Local Shop |
| Other | 4 | 4.6 | 39 | 35 | Tent |
| Total | 87 | 100 | 111 | 100 | Total |
| Size | | | | | |
| Small | 23 | 26.4 | 76 | 68.5 | Small |
| Medium | 49 | 56.3 | 29 | 26 | Medium |
| Large | 15 | 17.3 | 6 | 5.5 | Large |
| Total | 87 | 100 | 111 | 100 | Total |
| Operational year | | | | | |
| >5 years | 30 | 34.5 | 38 | 34.2 | >5 years |
| 5–10 years | 21 | 24.1 | 47 | 42.3 | 5–10 years |
| 10 years up | 36 | 41.4 | 26 | 23.5 | 10 years up |
| Total | 87 | 100 | 111 | 100 | Total |

Source: Questionnaire Survey, 2019.

Figure 6 shows that hotels, restaurants, resorts, and tents produced waste per day with not much difference (16, 14, 11, 10 kg/day, respectively). This meant that irrespective of the business size, they all produced waste that impacted the environment quite similarly.

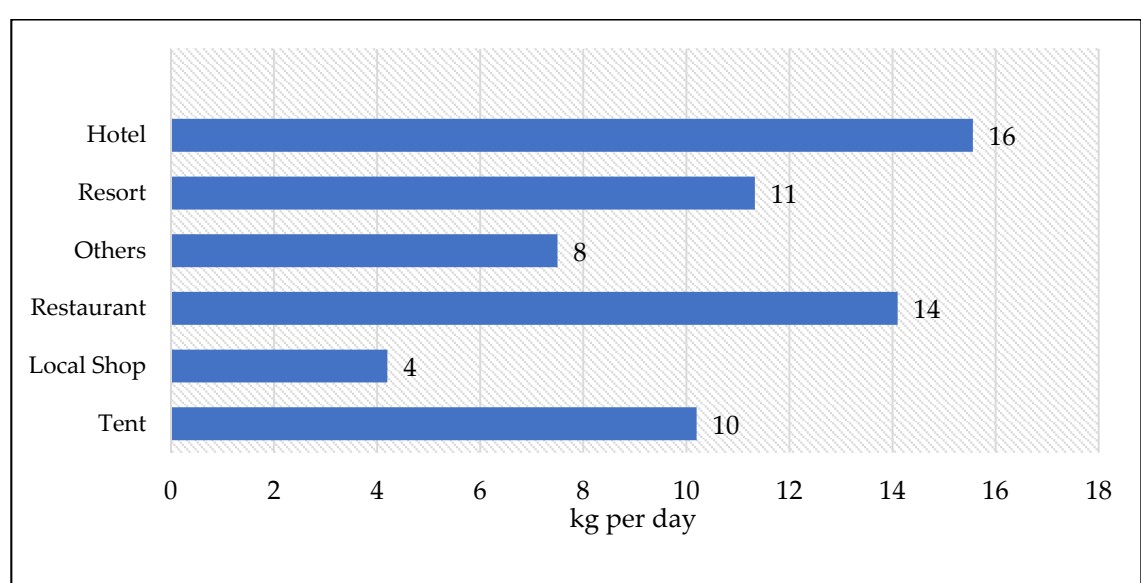

**Figure 6.** Average Waste Generated per Day per Private Sector Type.

From Table 6, it was found that the accommodation sector ranked the willingness of MSWM participation as "Absolutely agree" ($\overline{X}$ = 4.51 and S.D. = 0.73); with the first reason as 'the collaboration among communities, tourist business, and government for sustainable MSWM is important' as "Absolutely agree" ($\overline{X}$ = 4.57 and S.D. = 0.66); followed by 'the current waste management system should be improved' as "Absolutely agree" ($\overline{X}$ = 4.52 and S.D. = 0.71). Trades also ranked the willingness of MSWM participation as "Absolutely

agree" ($\overline{X}$ = 4.51 and S.D. = 0.92), with the first reason the same as the accommodation sector, but they were different for 'if there is a specific organization giving a formal role in the waste management system, you will join' at "Absolutely agree" ($\overline{X}$ = 4.47 and S.D. = 0.92).

**Table 6.** The Willingness to Participate in MSWM. n = 198.

| Attitude Statement | Statistics | | Opinion Level | Rank |
|---|---|---|---|---|
| | $\overline{X}$ | S.D. | | |
| Accommodations | | | | |
| - The current waste management system should be improved | 4.52 | 0.71 | Absolutely agree | 2 |
| - The collaboration among communities, tourist business, and government for sustainable MSWM is important | 4.57 | 0.66 | Absolutely agree | 1 |
| - If there is a reorganization of MSWM from the current system to a new sustainable collaborative system, you will join | 4.46 | 0.80 | Absolutely agree | 4 |
| - If there is a specific organization giving you a formal role in waste management the system, you will join | 4.51 | 0.75 | Absolutely agree | 3 |
| Total | 4.51 | 0.73 | Absolutely agree | |
| Trades | | | | |
| - The current waste management system should be improved | 4.58 | 0.97 | Agree | 1 |
| - The collaboration among communities, tourist business, and government for sustainable MSWM is important | 4.58 | 0.83 | Absolutely agree | 1 |
| - If there is a reorganization of MSWM from the current system to a new sustainable collaborative system, you will join | 4.43 | 0.98 | Agree | 3 |
| - If there is a specific organization giving you a formal role in waste management the system, you will join | 4.47 | 0.90 | Absolutely agree | 2 |
| Total | 4.51 | 0.92 | Absolutely agree | |

Source: Questionnaire Survey, 2019.

Table 7 shows that pollution reduction was the positive result; the sorting waste indicator was 80–97% of the total, while the MSWM service payment indicator was 50–70% of the total private sector respondents. The environmental protection aspect was quite a negative result; the importance to improve environmental performance was a concern to only 50%, while the environmental impact from their business was a concern for 39–50% of all private sector respondents. The decreased resource and carbon intensity aspect had a different result in the accommodation sector with the environmental policy indicator at 63% with trades only 26%, while the plan to decrease resource consumption indicator had a positive result for both private sector areas of 70–80% of the total. Finally, the supporting sustainable tourism aspect had a positive result for both respondents for the two indicators (minimizing waste and promoting sustainable tourism and nearby places) at 90–95%.

**Table 7.** Green Competency Evaluation of Private Sector. n = 198.

| Green Competency | Items | Accommodations | | Trades | |
|---|---|---|---|---|---|
| | | **F** | **%** | **F** | **%** |
| **Pollution Reduction** | | *Do you sort waste before disposing of it?* | | | |
| | Yes | 85 | 97.7 | 89 | 80.18 |
| | No | 2 | 2.3 | 22 | 19.82 |
| | Total | 87 | 100 | 111 | 100 |
| | | *Do you pay the municipal solid waste service fee?* | | | |
| | Yes | 63 | 72.4 | 55 | 49.55 |
| | No | 24 | 27.6 | 56 | 50.45 |
| | Total | 87 | 100 | 111 | 100 |
| **Environmental Protection** | | *Is it important to improve environmental performance?* | | | |
| | Yes | 46 | 52.8 | 60 | 54.05 |
| | No | 41 | 47.2 | 51 | 49.95 |
| | Total | 87 | 100 | 111 | 100 |
| | | *Are you concerned about the environmental impact of your business?* | | | |
| | Yes | 33 | 38 | 55 | 49.5 |
| | No | 54 | 62 | 56 | 50.5 |
| | Total | 87 | 100 | 111 | 100 |
| **Decreased Resource and Carbon Intensity** | | *Do you have an environmental policy?* | | | |
| | Yes | 55 | 63 | 28 | 26 |
| | No | 32 | 37 | 83 | 74 |
| | Total | 87 | 100 | 111 | 100 |
| | | *Have you planned to decrease resource consumption?* | | | |
| | Yes | 61 | 70 | 89 | 80 |
| | No | 26 | 30 | 22 | 20 |
| | Total | 87 | 100 | 111 | 100 |
| **Supporting Sustainable Tourism** | | *I always minimize the amount of waste from my business* | | | |
| | Yes | 82 | 95 | 104 | 94 |
| | No | 6 | 5 | 7 | 6 |
| | Total | 87 | 100 | 111 | 100 |
| | | *I always promote sustainable tourism and nearby attractive places* | | | |
| | Yes | 83 | 95.6 | 105 | 95 |
| | No | 4 | 4.4 | 6 | 5 |
| | Total | 87 | 100 | 111 | 100 |

Source: Questionnaire Survey, 2019.

Table 8 shows that there were six indicators of green competencies that were associated significantly with the private sector accommodation and trades groups at the acceptance of statistical significance 0.05. Furthermore, these six indicators tested the homogeneity to find out the similarity and differences in behavior of both groups. Finally, the test found that from the six associated indicators, there were only three indicators (sorting waste, improve environmental performance, and promoting sustainable tourism) that had similar behavior for each. Therefore, all the results from the survey of the private sector can be the change agent of MSWM in the coastal tourist destinations.

**Table 8.** Chi-square Test for Associated Green Competencies of Private Sector. n = 198.

| Green Competency | Indicators | $X^2$ | Sig. |
|---|---|---|---|
| Pollution Reduction | -Sorting Waste Before Disposing | 7.107 [a] | 0.008 |
| | -Waste Service Payment | 10.912 [a] | 0.001 |
| Environment Protection | -Improved Environmental Performance | 6.575 [a] | 0.014 |
| | -Environmental Concerning the Business's Impact | 0.167 [a] | 0.773 |
| Decreased Resource and Carbon Intensity | -Environmental Policy | 35.875 [a] | 0.000 |
| | -Decreases Resource Consumption | 19.656 [a] | 0.960 |
| Supporting Sustainable Tourism | -Minimized Waste | 10.747 [a] | 0.001 |
| | -Promoting Sustainable Tourism | 9.621 [a] | 0.001 |

[a]. 0 cells (0.0%) have an expected count of less than 5. The minimum expected count is 8.64. [b]. Computed only for a 2 × 2 table. Source: Questionnaire Survey, 2019.

At the end of the questionnaire, there was an open question about what are the needs of the private sector to improve the capacity of MSWM in their business. A summary of the challenges, potentials, and needs is in Table 9.

**Table 9.** Summary of Private Sector Exploration Results.

| | | Challenges | | Potentials | | Need |
|---|---|---|---|---|---|---|
| Common | PrC1 | Produced high waste similarly of both accommodation and trade | PrP1 | Having the same attitude toward effective stakeholders on MSWM services of both accommodation and trades. | PrN1 | Need both sectors to decrease the use of plastic bags. |
| | | | PrP2 | Having a high willingness for participation from both accommodations and trades. | PrN2 | Need employees in the private sector to carefully sort waste before disposing of it. |
| | | | PrP3 | Sorting waste, improve environmental performance, and promote sustainable tourism was an important indicator that was associated with all private sectors. | PrN3 | Need more collaboration between public and private. |
| Specific | PrC2 | Less concern on environmental impact from their business of trades group | PrP4 | Accommodations have an environmental policy and minimizing waste while trades do not have. | PrN4 | Need more following clean up, the separation waste truck, bigger size of bins, and the cover. |
| | | | | | PrN5 | Need to clean up waste along the beach and need a markeddecrease of waste. |

Note: PrC (Private Sector Challenges), PrP (Private Sector Potentials), PrN (Private Sector Needs).

Table 9 shows that the private sector, both accommodation and trades, have many potentials, and an overview of service characteristics of the private sector can support the economic growth and mass tourists flowing into the area because of their diversity of business types and operational years. Even though the accommodation and trade groups did not have greatly different results in waste generation per day, they still had the

potential for green competency in several aspects; pollution reduction, decreased resource and carbon intensity, and supporting sustainable tourism.

*4.3. Synergistic Opportunities*

This section analyzes the empirical result from the public sector investigation and private sector exploration by matching up the challenges, potentials, and needs for synergistic opportunities. By managing the complexity of the analysis findings for interpretation, the synergistic opportunities could be categorized into four groups based on the type of sectors and the synergy concepts. From the literature review, synergy could be found in the form of potential synergy or potentials and needs synergy to serve as leverage for effective, equitable, and sustainable programs for poverty eradication and growth [25]. However, this research found more opportunities which are the challenge synergy to solve the common problems and the new interest sector synergy, or the third sector who are not involved much in SWM but can improve the MSWM service in the coastal tourist sub-region.

4.3.1. Challenging Synergies of the Public Sector

Based on the specific challenges of municipalities without a PPP in the coastal tourism sub-region, they faced disadvantages in many aspects (PuC4-6: role, manpower, communication, landfill, and cost control). Thus, "Sharing Landfill" together intergovernmental for without a PPP cases would receive a multi-benefit and multi-layered relationship. It would produce both direct and indirect benefits to all collaborated municipalities without a PPP. Apart from solving those mentioned challenges, it also has a high power to attract private sector investment for this landfill sharing.

The requirement of municipalities with a PPP to negotiate with a private partner every contract year (PuC3) can be another opportunity to create synergy from this situation by "Expanding the MSWM Service Area" for power bargaining. This synergy could mean expansion in terms of geography or product when the acquiring organization demands more attached benefits like reputation or capital. Thus, expanding will help in increasing organizational debt capacity (PuC2). Apart from the success in negotiation, it could improve non-proper landfill (PuC7) in without a PPP cases as well.

4.3.2. Potential Synergies of the Private Sector

There is a common potential in the willingness of participation with a high sorting waste behavior from both accommodation and trade sectors (PrP1 and PrP2). While they all need extra activities along the beach, (PrN5) "Creating the Green Tourism Network and Activities" between accommodations and trades could reach a win-win situation. Building this network to offer eco-tourism activities for tourists/customers or offer tourism choices to people could make a positive impact on the environment and economy. On one hand, it would help to minimize waste and provide environmental protection from their synergized activities. On the other hand, it would support tourism by attracting a more specific type of tourist.

There were three indicators of green competency significantly associated with the accommodation and trade groups (PrP3). While there were some different behaviors between trades and accommodation related to their employees (PrP4 and PrN3). Thus, "Adding important behavior as the rule to continue the enterprise license" would have a positive impact on all areas of the private sector for synergizing further their operations to reach the rule defined. It is an economic environment that all sectors in coastal tourist destinations need to realize that their business must align with environmental protection for operational sustainability.

4.3.3. Crossing Synergies between Public and Private Sectors

Due to the private sector areas producing a similarly high volume of waste per day, both the public and private sectors have a mutual need for more collaboration to protecting

the local environment (PrN 3 and PuN5-6). Thus, these linkages could be "Supporting the Existing Green Group or New Enterprises Network". Municipalities should support the green enterprise group/network formulation, operation, and public relations for their activity. Support by the public sector becomes an impactful tool to gain trust from local people, tourists, and enterprises.

While accommodations and trades have less concern for the environmental impact from their business (PrC2), the public sector has a high potential for decentralized authority and willingness to collaborate with other sectors (PuP2-3). Thus, the public sector should provide a "Practical Method and Benefit of being Green Enterprises" to attract the private sector by the in-kind benefit (e.g., acknowledge, training, and educating about green enterprise) and in-cash benefit (i.e., the discount price of buying the recycling equipment and tax reduction).

4.3.4. Synergies with Other Types of Partnership

This section analyses were based on the findings that both public and private sectors had an expectation of contributing more to the management of waste in their territory and to enhancing their MSWM capacity.

There was a complexity of reporting back to the government on environmental performance (PuC1 and PuN8), and a need for more research on MSWM an increased awareness of people's conscious waste sorting behavior (PuN1-2). Thus, most municipalities in the coastal tourist sub-region should refer to the "Collaboration with Research Institution or University" for the initiatives from the data science center. It helps the public sector to update and create a systematic report of environmental performance on both offline and online platforms. Apart from preventing the complexity of performance reporting, the target organization can utilize the data source to increase their specialization in research and education.

Ineffective communication was evident with some areas communicating directly to every person, while others communicate through a group (PuC8). To solve these situations and attract greater stakeholder participation, such as applications for green enterprises labels or decreasing plastic use (PrN1), these cases need high impact communication to make a change (PuN3-4). Thus, both the public and private sectors should refer to have a "Partnering with NGO and CBO" as the power communication mechanism. It would help to launch many campaigns, spread the proper message, and prevent misunderstanding of MSWM services in the coastal tourist sub-region.

## 5. Discussions

This section discusses the derived-benefits from those analyzed synergistic opportunities to ensure that it leads to partnership effectiveness and sustainability [29]. Based on the literature review, to maximize synergy and keep the partners engaged, it needs to be efficient in the extent to which the roles and responsibilities of partners match their interests and skills and ensure the benefits derived from being in the partnership [30]. Thus, all analyzed partnership synergy as an output will be discussed to find out the benefit derived outcomes from working together.

### 5.1. The Benefits Derived from Partnership Synergy

In the previous study on the synergy of patient care [24], we found a lot of benefit-derived outcomes. It is true that different context studies have different results and benefit-derived findings. The empirical finding of these analyzed synergistic opportunities was different from the previous one and is highlighted as follows.

- Problems solved by the challenges synergistic opportunities of the public sector both in municipalities with and without a PPP such as workload, negotiation, and disadvantage on the organizational aspect.

- Green enterprises increased, from the potential synergistic opportunities of the private sector, they have a high potential to be a part of MSWM enhancement and a change agent of coastal tourist sub-region.
- Functionality changed, a part of focusing on customer service, the crossing synergistic opportunities between the private and public sectors will lead to more focus on environmental hospitality. These synergistic opportunities will bring a positive impact on both the environment and the economy.
- Multi-sector partnerships, the synergistic opportunity of the new interest sector being referred to other potentials organizations can have a hand in supporting the MSWM in the coastal tourist sub-region. It will evolve MSWM in the coastal tourist sub-region from being a PPP (3Ps) to a PPPP (4Ps) which includes the people sector.

The analyzed results of this research were based on the findings of current challenges, potentials, and needs of all sectors. It ensured that the synergistic output opportunities and the derived benefit outcome will respond to public and private sector sustainability [16]. A visualization of the overall partnership toward synergistic opportunities of MSWM in the coastal tourist sub-region is shown in Figure 7.

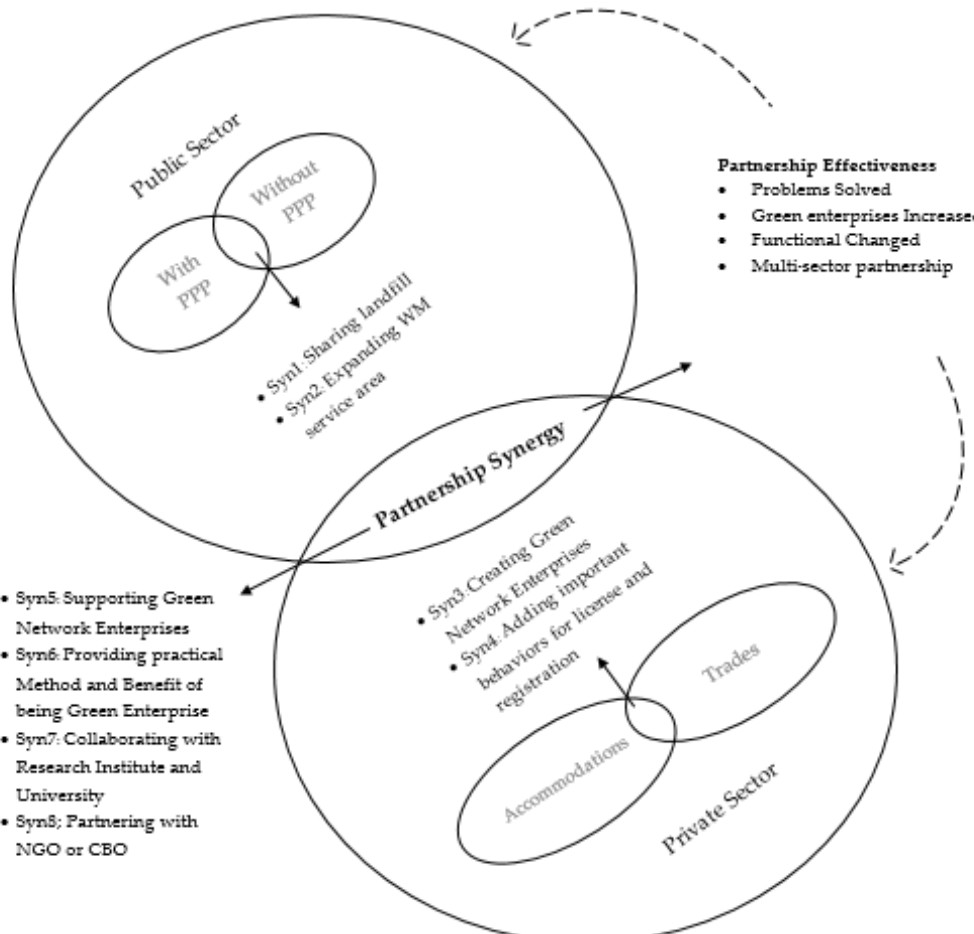

**Figure 7.** Analyzed Content with Framework Modification from [27].

### 5.2. Implications

As presented in the challenges of synergistic opportunities a synergistic partnership study is lacking. This study found that it is appropriate to add the synergy concepts as the ground theory to achieve the sustainability of MSWM in the coastal tourist sub-region. Enhancing sustainable MSWM needs a dynamic organizational transformation that consists of multi-sectors that have different resources and expertise to contribute to

the operation [21]. Analyzing the empirical results of this research also revealed that a multi-sector partnership would enhance the current MSWM services in the coastal tourist sub-region. All mentioned organizations in the empirical analyzed results were experts in their operation such as the NGOs, CBO, research institutions, and universities. In addition, the potential synergistic opportunities were the change agents that can make a dynamic movement in the development context.

As addressed, the synergy concept was complicated in context because of the different aspects between the services and ecology. Waste management in tourist destinations was the service for decreasing the environmental impact. It needs to find the synergistic opportunities to be exploited at the right time by the right people, towards specific targets as a potential synergy to reach a win-win situation [13]. In this sense, no matter the service or ecological context, the synergy concept will ultimately put the whole MSWM industry in an integrated relationship to reach the optimal partnership and obtain the benefits. Thus, applying the synergy concepts is explicable to enhance MSWM effectiveness and expand the collaborative governance for balancing both tourism and the environment.

## 6. Conclusions and Recommendations

The public sector along the coastal line has different MSWM strategies: with public-private partnership (PPP) and without PPP, with many challenges in the study areas of the EEC region, while the private sector has many potentials of MSWM. Both public and private sectors have synergistic opportunities in four aspects: (1) challenging synergies within the public sector by sharing landfill and expanding the MSWM area of the public sector, (2) potential synergies via the private sector by creating a green network enterprise and requiring appropriate behavior to register and maintain licenses for the private sector, (3) synergies with cross-sector partnerships by supporting a green network and providing practical policy of being a green label, and (4) synergies through other types of partnership for enhancing MSWM effectiveness. There were five benefit- derived outcomes of being in the synergy partnership above: (1) Problems solved, (2) Green enterprises increased, (3) Functional change, and (4) Multi-sector partnership. Therefore, it will be ensured that those analyzed synergistic opportunities will reach partnership effectiveness automatically and enhancement of MSWM services toward sustainability.

With respect to many available MSW techniques and tools to evaluate the quality of MSWM, the synergies toward partnership effectiveness of MSWM in the coastal tourism sub-region can fill the gap in research limitations as grounded theory. Exploring these multi-sector groups realized the existing challenges, potentials, and needs that can be useful in lessening the research limitations and applied to other places to shift MSWM services as a fundamental public service towards sustainability. The limitations of this research included the data privacy that prevented some private sector accommodation and trade groups to answer in the questionnaire survey. For future research, as there were many references to the people sector, there should be a study on the synergistic partnership that includes the people sector to improve the synergies of a multi-sector partnership towards improving MSWM services in coastal tourism destinations.

**Author Contributions:** Designed conceptualization and methodology, S.J. and V.N.; collected and analyzed the data and drafted the manuscript, S.J.; worked on the flow, organizational structure, discussions, conclusions, and review of the manuscript, S.J.; revised the manuscript and correspondence, V.N.; contributed to improving the clarity of the research and review and revising the manuscript, V.N., K.K., W.X. All authors have read and agreed to the published version of the manuscript.

**Funding:** This research was funded by the Asian Institute of Technology Fellowship.

**Acknowledgments:** All the respondents who give up the time to be interviewed and the questionnaire surveyed were gratefully acknowledged.

**Conflicts of Interest:** The authors declare no conflict of interest.

# Appendix A

**Questionnaire for Private Sector**

-----------------------------------------------------------------------------------------------------------------

Time:....................

**Part I: Respondent's information**

1.1 Private organization's name:...........................

1.2 Respondent's name....................... age............ (Years) Sex......

1.3 Position....................... Work for........ (Years)

1.4 Tel.......................

1.5 Email:......................

**Part II: Service Characteristics**

2.1 Your tourist business types:

☐ Hotel      ☐ Resort           ☐ Room rent

☐ Restaurant      ☐ Local shop/grocery      ☐ Tents ☐ Others....

2.2 If you are a hotel and resort, please identify the number of rooms.......................

2.3 Number of workers....................... (People)

2.4. The size of your business

☐ Small ☐ Medium      ☐ Large

2.5. Age of your business............ (Years)

2.6 Building material....................... (Such as cement, bamboo, wood or grass) If mixed material, can describe hotel material characteristic of material using here:

............................................................................................................................

............................................................................................................................

2.7 Number of tourists by day, month, and year (ask for data recorded)

............................................................................................................................

............................................................................................................................

2.8 Water supply and electricity per day, month, and year (ask for data recorded)

............................................................................................................................

............................................................................................................................

2.9 How much waste is generated from the accommodation (Kg/day, month, year) (Ask for data recorded)

............................................................................................................................

............................................................................................................................

**Part III: Attitude toward MSWM**

3.1. Do you think that collaboration among multi-sector (public-private-people) is important?

☐ Yes           ☐ No

3.2. The willingness of participation

| (1 = absolutely disagree, 5 = absolutely agree) 1 2 3 4 5 | 1 | 2 | 3 | 4 | 5 |
|---|---|---|---|---|---|
| The current waste management system should be improved | | | | | |
| The collaboration among communities, tourist business, and government for sustainable MSWM is important | | | | | |
| If there is a reorganization of MSWM from the current system to a new sustainable collaborative system, you will join | | | | | |
| If there is a specific organization giving you a formal role in the waste management system, you will join | | | | | |

**Figure A1.** *Cont*.

**Part IV: Green Competency**
**Pollution Reduction,**

4.1. Do your business sort waste into different types before disposing of?
　　　☐ Yes,　　　　　　　　☐ No *(go to question 2)*
4.2. How often do your business sort waste before disposing of?
　　　　☐ Every time　　　　　　　☐ Once a week
　　　　☐ Twice a week　　　☐ Three times a week
　　　　☐ Four times a week　　　　☐ Five times a week
　　　　☐ Others (please specify) _________________________
4.3. What types of waste that your business sort before disposing of? *(More than 1 answer is possible)*
　　　　☐ Paper　　　　　　☐ Plastic　　　　　　☐ Food waste
　　　　☐ Glass　　　　　　☐ Metal　　　　　　☐ Beverage/food cans
　　　　☐ Mixed waste　　　☐ Toxic waste (light bulbs, batteries, used oil)
4.4. Do you have to pay for waste collection services?
　　　　☐ Yes, _________ baht　☐ No

**Environmental Protection,**

| 4.5 | Is it important for your business to improve your environmental performance?<br>　　☐　Yes<br>　　☐　No |
|---|---|
| | Are you concerned about the impact of your business on the environment?<br>　　☐　Yes<br>　　☐　No |

| 4.6 | Have you done anything to make sure that your tourism business is more environmentally friendly?<br>　　☐　Yes<br>　　☐　No |
|---|---|
| | If you answered 'yes' to the above, what have you done to ensure your tourism business is more environmentally friendly? (please specify) |

**Decreased Resource and Carbon Intensity**

| 4.7 | Does your business have an environmental policy?<br>　　☐　Yes<br>　　☐　No |
|---|---|
| | If No, would you consider developing one?<br>　　☐　Yes<br>　　☐　No |

| 4.8 | Have you planned to decrease resource consumption??<br>　　☐　Yes<br>　　☐　No |
|---|---|
| | If No, would you consider developing one?<br>　　☐　Yes<br>　　☐　No |

**Figure A1.** *Cont.*

**Supporting Sustainable Tourism**

| 4.9 | Please let us know whether you agree or disagree with the following statements:<br>a) I always try to minimize the amount of waste my business produces<br>    ☐   Agree<br>    ☐   Disagree<br><br>b) Wherever possible I promote nearby attractions and facilities<br>    ☐   Agree<br>    ☐   Disagree |
|---|---|

**Part V: The Needs on Waste Management of Private Sector**

5.1. What are the needs of your organization's waste management?

.......................................................................................................................

.......................................................................................................................

5.2 What are the needs of municipal waste management in a tourist destination?

.......................................................................................................................

.......................................................................................................................

**Figure A1.** Questionnaire for for private sector.

**Table A10.** Checklist Question for Semi-Structured Interview with Municipalities.

| Policy Aspects | |
|---|---|
| Relevant Policies,<br>Strategies on MSWM | 1. What are the policies decentralized to your organization?<br>2. Are the policies and legalization sufficient;<br>3. Is the current MSWM operation accord to the rule and regulation defined? |
| **Organizational Aspect** | |
| Roles and Responsibilities<br>Capacity and Labor Tenure | 1. What are the main roles of the municipal waste management system?<br>2. How do you monitor and control MSWM?<br>3. Is your skilled-staff enough and can work legally? |
| **Technical Aspect** | |
| Seasonal Waste<br>Collection Rate and Efficiency | 1. The amount of waste high & low seasons<br>2. Can it handle waste in both high and low seasons?<br>3. The percentage of collected waste in both high and low seasons<br>4. Is the management waste system enough for the population number? |
| **Social Aspect** | |
| Stakeholders' participation<br>Communication mechanisms | 1. Do tourists and the population believe the SWM is a benefit to society?<br>2. Do they ready to support the MSWM? If yes, please give some examples of the situation or project from the people sector!<br>3. How to communicate among municipality-contracted partner-people for SWM? |
| **Economic Aspect** | |
| Fiscal viability and Trend<br>Tax and Willingness to corporate | 1. The arrangement of the municipality's revenue, expenditure, how much portion between MSWM expenditure and the total expense<br>2. The trend of MSWM expenditure<br>3. How much for the tourism zone? how much for the community zone?<br>4. Are they willing to pay MSWM tax? |
| **Environmental Aspect** | |
| Cleanliness<br>Health Impact | 1. Is there any improper disposal of the area?<br>2. Is the current waste collection clean enough?<br>4. People in the community have good health? |

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
