# Peer review of "Partnership towards Synergistic Municipal Solid Waste Management Services in a Coastal Tourism Sub-Region"

_sustainability, doi:10.3390/su13010397_

Round 1
Reviewer 1 Report
I think the manuscript is well written and it is technically sound but however, there are few issues that need attention;
- The sample size is specified in the manuscript as well as the rationale for the sampling?
- There is a need to clarify hoe significantly are the difference between low season and high season?
- Line 345-348 needs to be clarified as the figure does not depict what the authors are saying?
- Authors need to provide the questionnaire for a better understanding of the manuscript.
- The language in the manuscript should properly be edited.
Author Response
Please see the attachment
But this file has combined the revision from two reviewers, it may looks too much revising apart from your comments.
Apologize for late submission

Reviewer 2 Report
Abbreviations like SWM are not used consistently, eg (MSW)
The applicability of the Synergy Model requires further justification. A literature review of the Synergy Conceptual Modelling as applied outside the traditional patient nursing scope would add value.
Figure 5 Conceptual Framework as an adaptation of [25], [27], [29] requires justification of the type of adaptation applied and why.
The interview process is unclear. Likewise is the methodology of the analysis of the interview outcomes.
Author Response

(The authors gave the same response as above.)
